# Effect of Cooling Rate after Solution Treatment on Subsequent Phase Separation Evolution in Super Duplex Stainless Steel 25Cr-7Ni (wt.%)

**Jianling Liu** [1,*] , **Yadunandan Das** [1,†] , **Stephen M. King** [2] , **Jan Y. Jonsson** [3], **Sten Wessman** [4] **and Peter Hedström** [1]

1    Department of Material Science and Engineering, KTH Royal Institute of Technology,
     SE-100 44 Stockholm, Sweden; ydas@kth.se (Y.D.); pheds@kth.se (P.H.)
2    ISIS Pulsed Neutron and Muon Source, STFC Rutherford Appleton Laboratory, Harwell Campus,
     Didcot OX11 0QX, UK; stephen.king@stfc.ac.uk
3    R&D Avesta, Outokumpu Stainless AB, P.O. Box 74, SE-774 22 Avesta, Sweden;
     jan.y.jonsson@outokumpu.com
4    Swerim AB, P.O. Box 7047, SE-164 07 Kista, Sweden; sten.wessman@swerim.se
*    Correspondence: jianling@kth.se; Tel.: +46-(0)72-920-8032
†    Current address: Materials Engineering, The Open University, Walton Hall, Milton Keynes MK7 6AA, UK.

**Abstract:** The effect of cooling rate after solution treatment on the initial structure of super duplex stainless steel 25Cr-7Ni (wt.%), and the effect of the initial structure on phase separation (PS) evolution during subsequent aging were investigated. The nanostructure in the bulk of the steel was studied using small-angle neutron scattering (SANS). Ex situ SANS experiments showed that the rate of PS differs during aging, due to the different initial structures imposed by the difference in cooling rate after solution treatment. In situ SANS experiments revealed that the PS is already pronounced after aging at 475 °C for 180 min and that a slower cooling rate after solution treatment will lead to more significant PS. Hence, PS depends on the plate thickness, imposing different cooling rates in the production of duplex stainless steels.

**Keywords:** cooling rate; phase separation; spinodal decomposition; super duplex stainless steel; small-angle neutron scattering; 475 °C embrittlement

## 1. Introduction

Super duplex stainless steels (SDSSs) are of strategic importance for critical components in, for example, the chemical industry, the off-shore industry and nuclear power plants due to their combination of high strength and corrosion resistance [1]. However, they suffer from the notorious '475 °C embrittlement' phenomenon, causing increased hardness and decreased toughness, if they are subjected to service temperatures above about 250 °C for prolonged periods [2]. The embrittlement is due to the decomposition of the body-centred cubic (bcc) ferrite phase into a Fe-rich bcc phase ($\alpha$) and a Cr-rich bcc phase ($\alpha'$) via spinodal decomposition (SD) or nucleation and growth (NG) [3].

The above-mentioned phase decomposition, or phase separation (PS), occurs on the nanoscale in a temperature range of about 275–500 °C for SDSS [4,5]. It is known that the ferrite in the duplex stainless steel (DSS) can already experience PS after solution treatment [6–8], depending on the solution treatment temperature [9–11] and the cooling rate after solution treatment [12–16]. Early simulation works [12,13] studied SD in binary Fe-Cr based alloys using the linearised Cahn–Hilliard model, and showed that a slower cooling rate would lead to more pronounced PS during continuous cooling, but if the cooling rate exceeded a critical value then PS could be avoided. Later, it was experimentally observed by Hedin et al. [14] using atom probe tomography (APT) that the initial Cr concentration amplitude was larger in unaged DSSs that had been exposed to a slower

cooling rate. The initial PS during cooling has been corroborated by Lemoine et al. [15], who showed that the ferrite hardness of unaged DSS exposed to a slower cooling rate after solution treatment was higher. Furthermore, it has also been shown that the cooling rate after solution treatment can affect the kinetics of PS during subsequent aging of binary Fe-Cr alloys, see e.g., Xu et al. [16].

As portrayed in the literature survey above, the heat treatment during production, in particular the cooling rate, is an important factor for predicting the evolution of PS during service. This understanding becomes particularly important for thick gauge sections of SDSS where it is not possible to cool the whole gauge thickness equally after solution treatment. In a recent study [17], it was shown that PS already occurs in the ferrite of as-fabricated thick hot-rolled SDSS. This, therefore, raises the concern how the gauge thickness influences the PS during heat treatment and subsequent aging during service. Hence, in the present work we set out to simulate the industrial production of thick and thin plates by applying two different cooling rates after solution treatment (slow and fast). The samples were investigated, both after the solution treatment and after further isothermal aging, using small-angle neutron scattering (SANS), an effective tool to probe the evolution of nanoscale structure in Fe-Cr-based alloys [5,18,19]. We describe the effect of cooling rate after solution treatment on the initial structure of the ferrite, and discuss the effect of the initial structure on the rate of PS during subsequent aging.

## 2. Materials and Methods

### 2.1. Material and Heat Treatment

A 6 mm-thick plate of hot-rolled super duplex stainless steel 25Cr-7Ni (SDSS 2507) was delivered by Outokumpu Stainless AB (Avesta, Sweden), the chemical composition of which is listed in Table 1. The representative microstructure of the SDSS 2507 is shown in Figure 1a, where the ferrite (dark) and austenite (bright) regions are seen to be elongated in the rolling direction. Samples of size 55 mm × 10 mm × 5 mm were heat treated using a Gleeble 3800 physical simulator (Gleeble, Poestenkill, NY, USA), where a combination of Ohmic heating and argon-gas quenching was applied in response to programmed cycles. A thermocouple was attached at the centre of each sample to control the thermal cycles. The cooling, after solution treatment at 1120° C, was performed at two different rates: slow and fast cooling, see Figure 1b. The cooling rates, shown in Table 2, were designed to simulate the cooling rate in the centre of an 80 mm plate (slow cooling) and a 20 mm plate (fast cooling), respectively. The slow-cooled (SC) and fast-cooled (FC) samples were both argon-gas cooled down to room temperature. Accelerated aging was conducted at 475 °C for up to 500 h (hereafter unaged samples are represented by 0 h aging time).

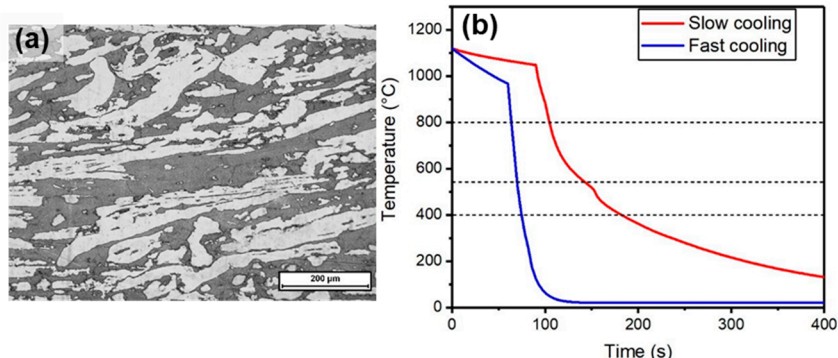

**Figure 1.** (**a**) Light optical micrograph of SDSS 2507 after etching using a modified Beraha II solution (100 mL water, 50 mL HCl, and 0.5 g $K_2S_2O_5$): the scale bar is 200 μm; (**b**) the cooling cycles imposed on the two different samples prior to ageing.

**Table 1.** Chemical composition of the 25Cr-7Ni super duplex stainless steel (wt.%).

| Material | Fe | C | Si | Mn | P | Cr | Ni | Mo | Cu | N |
|---|---|---|---|---|---|---|---|---|---|---|
| SDSS 2507 | Bal. | 0.012 | 0.30 | 0.83 | 0.023 | 24.84 | 6.90 | 3.80 | 0.18 | 0.28 |

**Table 2.** Cooling rates of the investigated samples at different temperature ranges.

| Samples | Cooling Rate (800–550 °C) | Cooling Rate (550–400 °C) | Cooling Rate (400–25 °C) |
|---|---|---|---|
| SC | 6 °C/s | 3.5 °C/s | 0.5 °C/s |
| FC | 40 °C/s | 30 °C/s | 5 °C/s |

### 2.2. Small-Angle Neutron Scattering (SANS)

SANS experiments were conducted on the SANS2d beamline at the ISIS Pulsed Neutron and Muon Source, Didcot, UK. This is a second-generation, white-beam, time-of-flight instrument where neutrons with incident wavelengths ($\lambda$) between 1.75 and 16.5 Å enable scattering vectors $Q = (4\pi \sin\theta)/\lambda$, where $2\theta$ is the scattering angle) in the range 0.002–1.5 Å$^{-1}$ to be measured simultaneously, depending on the actual configuration of the instrument [20]. The scattered neutrons were recorded on two large area, position-sensitive gas detectors of in-house design. For this work, the sample position was approximately 19 m from the moderator and the detectors were positioned 2.4 m and 4 m behind the sample, offset horizontally by −980 mm and +100 mm, respectively. The front detector was also rotated by 20 degrees into normal incidence. Steel samples for the SANS measurements were cut to dimensions of $10 \times 10 \times 1.1$ mm$^3$ and prior to the experiments were also ground and polished to remove any surface oxide film. The neutron beam was incident normal to the large face of each sample and collimated to a diameter of 6 mm.

Two series of measurements were undertaken: on samples aged ex situ for up to 500 h, and samples aged in situ for up to 3 h to monitor the evolution of PS in the early stages. All ex situ measurements were conducted at ambient temperature but inside an electromagnet (Goudsmit B.V., Waalre, The Netherlands) able to apply a magnetic field of up to 1.5 T across the samples orthogonally to the neutron beam, see Figure 2a. Previous research has shown that such an applied magnetic field can magnetically saturate the SDSS [5] and thereby permits a separation of the magnetic and nuclear contributions to the scattering that arise in steels [21]. Measurement durations were around 1 h. The in situ measurements were performed isothermally at 475 °C using a furnace of in-house design. However, it should be noted that all the in situ measurements were performed without an applied magnetic field, as this can influence the kinetics of PS [22].

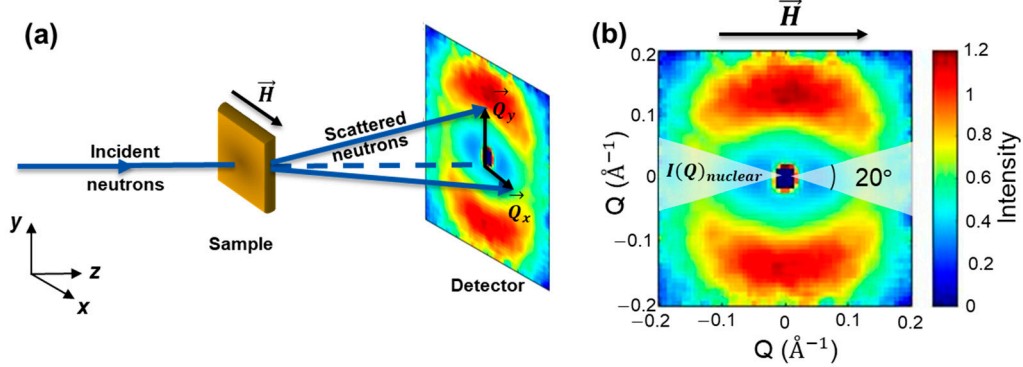

**Figure 2.** (**a**) Schematic of SANS technique; (**b**) illustration of the separation of the nuclear scattering obtained from 20° sectors ($\Psi = -10°$ to $+10°$) parallel to the magnetic field (applied along the *x*-axis).

Mantid Workbench (version 6.2.0, The Mantid Project, Mantid, Atlanta, GA, USA) [23,24] was used to azimuthally integrate and reduce the as-collected (raw) 2D SANS data. During this process, the raw data were corrected for factors such as the efficiency and spatial linearity of the detectors, the instrumental background scattering, the sample transmission, and the illuminated gauge volume. Due to the presence of a prominent Bragg edge in the sample transmission data, it was also necessary to exclude neutrons with wavelengths less than 4.3 Å from the data reduction, at the penalty of reducing the maximum accessible $Q$ to ~0.7 Å$^{-1}$ (this includes an additional reduction arising from a more limited angular range subtended from the magnet). Data reduction yielded the macroscopic coherent elastic differential scattering cross section ($d\Sigma(Q)/d\Omega$), colloquially referred to as the intensity, $I$ (unit: cm$^{-1}$) [20], as a function of scattering vector $Q$. These data were then put on an absolute intensity scale by scaling them to the scattering from a partially deuterated polymer blend of known molecular weight measured with the same instrument configuration [25]. For the ex situ SANS measurements in a saturating magnetic field, the total scattering $I(Q)_{total}$ is a function of the angle $\Psi$ between $Q$ and the direction of the applied field $H$:

$$I(Q)_{total} = I(Q)_{nuclear} + I(Q, H)_{magnetic} \, Sin^2\Psi, \tag{1}$$

where $I(Q)_{nuclear}$ and $I(Q)_{magnetic}$ are the contributions arising from nuclear scattering and magnetic scattering, respectively. In this work, $I(Q)_{nuclear}$, which contains the structural information of interest, were obtained by integrating the scattering in 20° sectors ($\Psi = -10°$ to $+10°$) about the detector equator, see Figure 2b. This yielded data of good statistical quality for subsequent analysis with only a ~2% magnetic scattering contribution included [21].

The reduced SANS data were analysed by a now established method [19], wherein a combination of a power-law background function [10,26], predominantly describing the surface scattering from the precipitates, and a generic function describing the scattering from PS (here assumed to arise due to SD), were model-fit to the scattering data [27] using the SasView software (version 4.2.2, The SasView Collaboration) [28]:

$$I(Q)_{fit} = I(Q)_{spinodal} + B_{fit}, \tag{2}$$

where $B_{fit} = A_{power\_law}Q^{-n} + B_g$, the prefactor $A_{power\_law}$ determines the relative contribution of that term, $n$ is the power-law exponent ($1.5 \leq n \leq 4$) and $B_g$ is the residual $Q$-independent background intensity. A spinodal model is an appropriate assumption here as the chemical composition of SDSS 2507 is located very close to, or within, the spinodal line of the Fe-Cr miscibility gap (MG) [29]. Furthermore, it has previously also been found that DSS experiencing PS in the transient region can be modelled assuming a spinodal-like nanostructure [29]. The spinodal function [27] is:

$$I(Q)_{spinodal} = I_{peak}\left(1 + \frac{\gamma}{2}\right)x^2 / \left(\frac{\gamma}{2} + x^{2+\gamma}\right), \tag{3}$$

where $x = Q/Q_{peak}$, $Q_{peak}$ is the position of the spinodal peak in the scattering and $I_{peak}$ is the intensity at $Q_{peak}$. The term $\gamma$ is equal to $d + 1$ or $2d$, depending on the nature of the interface, where $d$ is the dimensionality ($d = 3$). Here $\gamma = 6$ was used in this work [19]. The fit of Equation (2) to SANS data from systems exhibiting PS has been shown to be reasonably good [5,18,19]. However, to facilitate inter-sample comparisons in graphs, it is helpful to remove the background scattering prior to plotting. Our procedure for doing this yields what we refer to as the Normalized Scattering Intensity (NSI), a proxy for the underlying structure factor: first, the $I(Q)_{fit}$ is normalised by the background function $B_{fit}$ (i.e., we compute $\frac{I(Q)_{fit}}{B_{fit}}$) to reveal the shape of the underlying spinodal signal ($I_n$). This $I_n$ is then fitted to a Gaussian function, $S(Q)_{fit}$, as an approximation for the structure factor. The signals of PS were then obtained by removing the contribution of the background and

normalising it with respect to $B_{fit}$, i.e., $NSI(Q) = B_{fit}\left(S(Q)_{fit} - B_{fit}/B_{fit}\right)$. More details are described in [18,19,21].

The wavelength ($\Lambda$) of SD can be obtained from $Q_{peak}$ using the simple expression:

$$\Lambda = 2\pi/Q_{peak}, \tag{4}$$

In this experiment, PS occurs in the ferrite and contributes to the correlation peak in the SANS data. For the investigated SDSS 2507, the volume fraction of the ferrite phase was 40.6% [18]. If a phase-separated domain is assumed to be a cube of edge length $\Lambda/2$, then the amplitude can be estimated by [19]:

$$I_{peak} = \varphi \left(1 - \varphi\right) \left(\frac{\Lambda}{2}\right)^3 (\Delta\rho)^2, \tag{5}$$

where $\varphi$ is the volume fraction of the $\alpha'$ phase in the ferrite and $\Delta\rho$ is the difference in neutron scattering length density between the $\alpha$ and $\alpha'$ phases ($0.5 \times 10^{-6}$ Å$^{-2}$ for every 10 at.%). The amplitude ($A$ at.%) can then be obtained by the following expression [19]:

$$A \text{ (at.\%)} = \Delta\rho \frac{10 \text{ at.\%}}{0.5 \times 10^{-6} \text{ Å}^{-2}}, \tag{6}$$

This methodology was recently successfully applied to a study of the evolution of the amplitude and wavelength due to PS in DSSs [5].

## 3. Results

### 3.1. Ex Situ SANS

Figure 3 shows the SANS data (Figure 3a–c) and the NSI (Figure 3d–f) of SDSS 2507 subjected to different cooling rates after solution treatment, and then aging for different times at 475 °C. As can be seen in Figure 3a, the SANS data of the unaged samples showed minimal differences between the two different conditions. After a steady decrease in intensity in the low-$Q$ range, the SANS data of the unaged samples converge beyond $Q = 0.1$ Å$^{-1}$ into a very broad and weak hump, hinting that a very low level of nanostructural segregation is already present before ageing. This is more easily seen in the corresponding NSI of the unaged samples shown in Figure 3d. The peak position for the SC sample is slightly to the right of the FC samples and the peak intensity is higher, indicating a smaller length-scale (wavelength) of decomposition but larger amplitude of decomposition. The quantification of the wavelength and amplitude for all the different conditions is tabulated in Table 3. Although the values are small, this analysis clearly indicates that PS was initiated in all the samples during the solution treatment and cooling process prior to the aging.

The aged samples show more obvious signs of nanostructure development. Correlation peaks evolve in intensity and move to lower $Q$ as PS occurs. After 20 h aging, the intensity of the SANS peaks from the two samples show some differences, see Figure 3b, which can be seen more clearly in Figure 3e and Table 3. The wavelength of the two samples has increased, compared to the unaged case, and there is a significant increase in both the NSI and amplitude, particularly for the FC sample, indicating PS is more severe. In contrast to the unaged case, the SANS peak intensity, NSI, and amplitude of the SC sample are now the lowest of the three samples. At the same time, the wavelength of the two samples has become more similar, which means the size of $\alpha'$ precipitates in these samples after 20 h ageing is also similar. After 500 h aging, the SANS peaks of the two samples have noticeably shifted to lower $Q$ and the peak intensity has increased further, see Figure 3c,f. However, whilst this did translate into an obvious increase in wavelength, the amplitude only increased slightly compared to 20 h ageing, and it is now the SC sample which shows the greatest amplitude. This latter observation correlates with that of Xu et al. [16] who showed that a slower cooling rate led to more PS in Fe-Cr binary alloys, regardless of the

nominal Cr concentration, when aged up to 300 h. This means the initial structure still plays an important role in affecting the PS until 500 h aging.

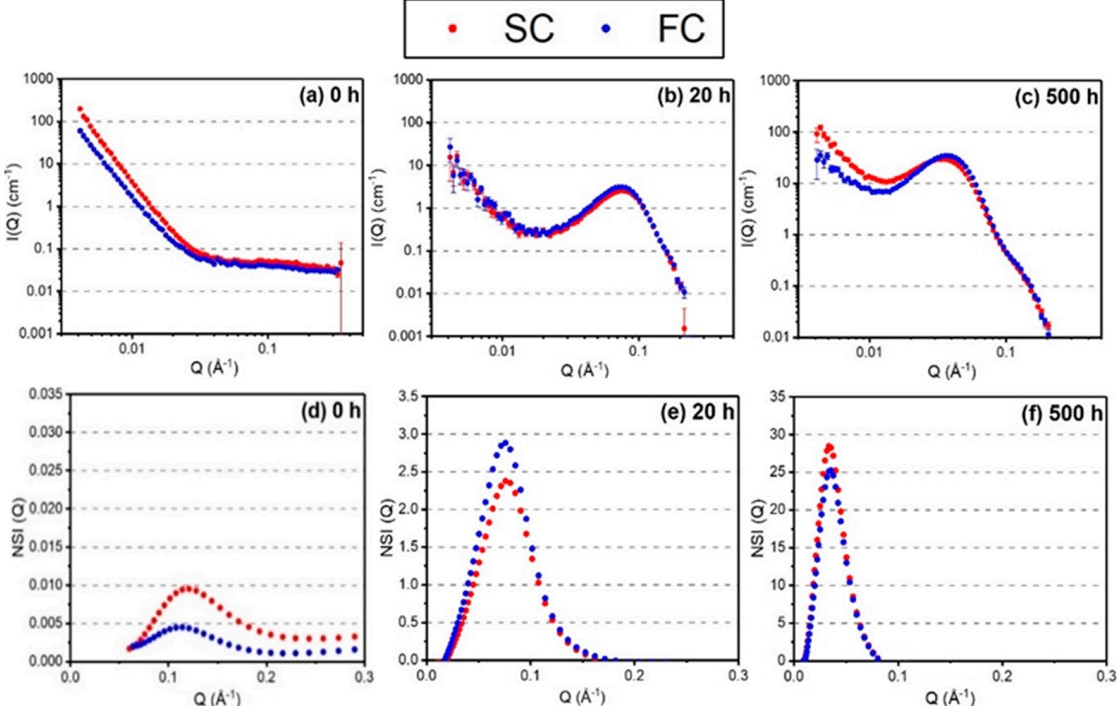

**Figure 3.** SANS intensity (**a–c**) and NSI (**d–f**) of SDSS 2507 ex situ aged at 475 °C (note that the *y*-axes of (**d–f**) are different).

**Table 3.** Spinodal wavelength and phase separation amplitude of the samples aged ex situ at 475 °C.

| Aging Time (h) | Wavelength (nm) | | Amplitude (at.%) | |
|---|---|---|---|---|
| | SC | FC | SC | FC |
| 0 | 5.6 ± 0.16 | 5.8 ± 0.20 | 7.7 ± 0.27 | 5.4 ± 0.51 |
| 20 | 8.2 ± 0.01 | 8.4 ± 0.01 | 59.9 ± 0.09 | 62.6 ± 0.08 |
| 500 | 17.2 ± 0.01 | 16.9 ± 0.01 | 67.6 ± 0.06 | 65.7 ± 0.06 |

### 3.2. In Situ SANS

Although the ex situ SANS data showed the effect of cooling rate during aging, the trend is somewhat inconclusive. From Table 3, it can be seen that the amplitude increases significantly during 0–20 h, where the embrittlement occurs [5]. In order to further analyse the early stage of PS, in situ SANS data are shown in Figure 4a,b. In the region $0.008 < Q < 0.04$ Å$^{-1}$ there is a feature which could be due to some oxide inclusions [16]. The length-scale is not interfering with the PS, and therefore, the feature is simply neglected in the background fitting [18]. This feature does not appear in the ex situ data, which could be because the scattering from the low-$Q$ region was suppressed by the 1.5 T magnetic field. In the high-$Q$ region ($0.04 < Q < 0.3$ Å$^{-1}$), the correlation peak increases with aging time for all samples. The evolution of PS can be seen more directly in Figure 4c,d. Since there is no significant change in wavelength, it is reasonable to assume that the PS was still at the early stage. However, it is noteworthy that the wavelength decreased slightly during the first 30 min compared to the unaged case, and then increased during subsequent aging. It is known that the critical wavelength decreases with the lowering of temperature within the MG [30]. Here, cooling from high temperature would keep the 'memory' of the initial structure with larger wavelength. In such case, during subsequent aging at 475 °C, the wavelength decreases first and then starts to grow. In the

first 180 min of aging, the FC sample shows a lower NSI and amplitude for a given aging time, compared to the SC sample. As can be seen in Table 4, the SC sample possesses a longer wavelength at each ageing time. Comparing the amplitude value to that of in Ref. [5], the PS in the first 180 min is shown to be already pronounced. The amplitude of the sample aged at 475 °C for 180 min is close to that of samples aged at 350 °C for 6000 h in Ref. [5], where the embrittlement is severe. In general, the SC sample shows more severe PS than the FC sample in the first 180 min, as shown by the larger $\alpha'$ size and compositional fluctuations. It is clear from these results that the cooling rate plays an important role in affecting the kinetics of PS at the early stage.

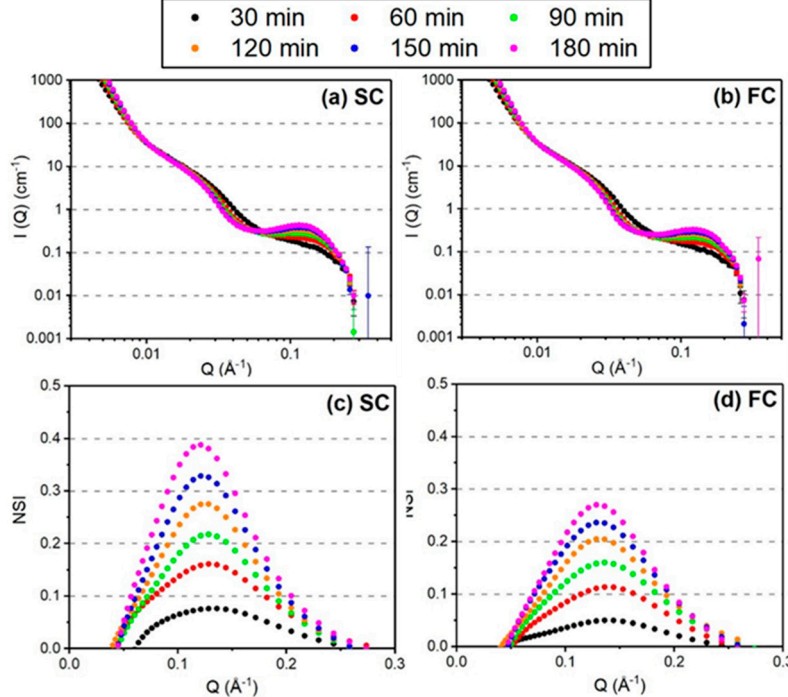

**Figure 4.** SANS intensity (**a**,**b**) and NSI (**c**,**d**) of SDSS 2507 aged in situ at 475 °C.

**Table 4.** Spinodal wavelength and phase separation amplitude of the samples aged in situ at 475 °C.

| Aging Time (min) | Wavelength (nm) | | Amplitude (at.%) | |
|---|---|---|---|---|
| | SC | FC | SC | FC |
| 30 | 4.1 ± 0.08 | 4.0 ± 0.16 | 26.9 ± 0.95 | 23.1 ± 1.37 |
| 60 | 4.5 ± 0.04 | 4.4 ± 0.08 | 33.3 ± 0.60 | 29.6 ± 0.84 |
| 90 | 4.8 ± 0.02 | 4.7 ± 0.03 | 39.2 ± 0.43 | 35.4 ± 0.36 |
| 120 | 4.9 ± 0.02 | 4.7 ± 0.03 | 42.9 ± 0.47 | 39.4 ± 0.37 |
| 150 | 5.0 ± 0.01 | 4.8 ± 0.02 | 46.2 ± 0.36 | 42.7 ± 0.45 |
| 180 | 5.2 ± 0.01 | 4.9 ± 0.02 | 48.1 ± 0.24 | 45.4 ± 0.30 |

### 3.3. Power-Law Fitting

The evolution of $\Lambda$ and $I_{peak}$ with aging time have been described by power laws in previous theoretical models, namely $\Lambda \propto t^a$ and $I_{peak} \propto t^b$ [16,31,32]. As can be seen in Figure 5a,b, there were obvious increases of $\Lambda$ and $I_{peak}$ in the two samples for up to 500 h. The values of the exponent $a$ in Table 5 are in general agreement with Hyde et al. [33], who reported a time exponent of ~0.25 for Fe-24Cr alloys aged at 500 °C for 500 h. The SC samples showed lower values of $a$ (0.22) but higher value of $b$ (0.79). However, it should be noted that the $a$ values here are all below 1/3 which means PS in the investigated samples has not reached the coarsening stage (where $\Lambda \propto t^{1/3}$) [34–36], although the wavelength increased significantly (refer to Table 3). The variation of $b$ shows the different degree of

PS in the two samples after extended aging, where SC shows a larger *b* value than FC. It is known that the initial difference due to cooling rate will be diminished if PS is in the final stages [14]. Combined with the SANS results, however, it is shown that differences due to the cooling rate are still obvious even when aged for 500 h. This corroborates the conclusion above that PS has not reached the late stage. For the first 3 h (see Table 5), it is seen that the exponent *a* is lower but *b* is higher, compared to the whole range of aging time (500 h). The fitting results for the first 3 h correlate with the quantification results from the in situ SANS data in Table 4, where the SC shows more pronounced PS than FC.

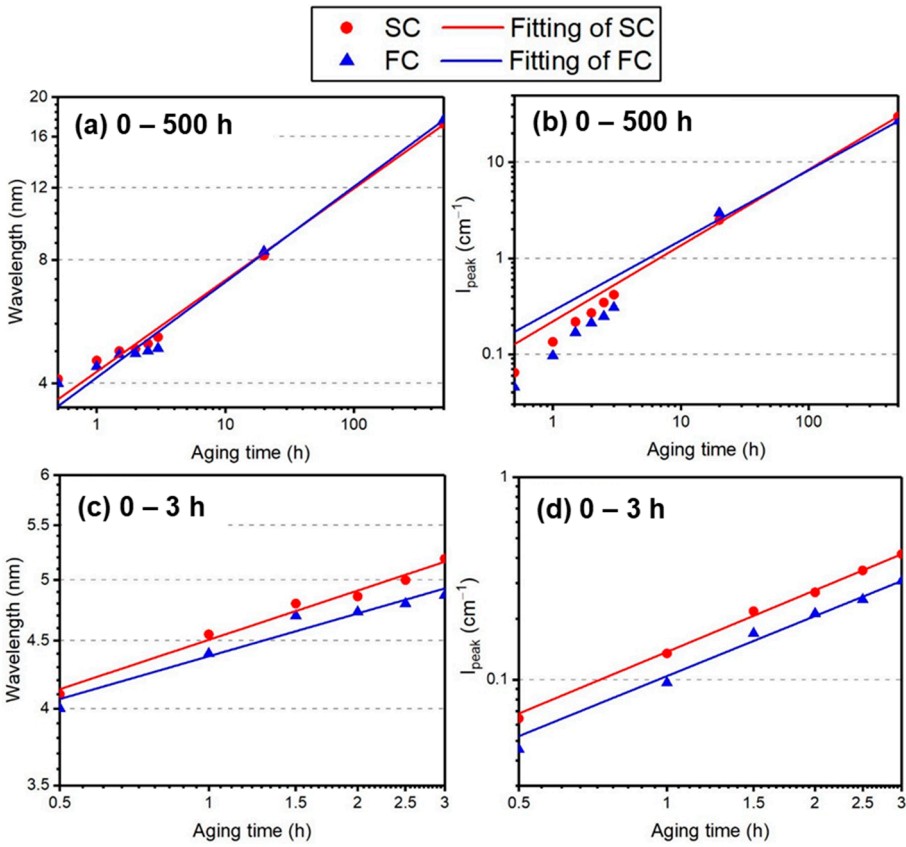

**Figure 5.** Spinodal wavelength (**a**,**c**) and peak intensity (**b**,**d**) as a function of the aging time (the straight lines are power-law fits from linear regression).

**Table 5.** Power-law exponents *a* and *b*, and linear regression coefficients $R^2$ , from fitting.

| Scheme | Exponent *a* | $R^2$ | Exponent *b* | $R^2$ |
|---|---|---|---|---|
| SC in (a) | 0.22 | 0.99 | 0.79 | 0.99 |
| FC in (b) | 0.23 | 0.99 | 0.73 | 0.99 |
| SC in (c) | 0.12 | 0.98 | 1.01 | 0.99 |
| FC in (d) | 0.10 | 0.95 | 0.98 | 0.99 |

## 4. Discussion

The rate of cooling from the solution temperature affected the initial nanostructure of the DSS. From the results presented here, and previous work [5], it is clear that a minor amount of decomposition already occurs in the as-cooled SDSS 2507, where the slower cooling rate leads to a larger Cr amplitude in the initial structure. This is because Cr demixing would be more pronounced for a slower cooling rate due to the longer time spent within the MG where there is a driving force to demix Fe and Cr [7,12]. For the fast cooling, the time spent inside the MG is shorter so the initial PS is less pronounced.

This initial difference caused by the cooling rate also affects the subsequent aging kinetics. The SC initial structure experienced a faster rate of PS in the early stages, which is due to the initial nuclei of $\alpha'$ that form during slow cooling and which provides a precursor for the subsequent PS. These nuclei grow and coarsen during the isothermal aging, which is mainly diffusion-controlled. In comparison, such a high degree of initial nucleation of $\alpha'$ would be less prominent during the fast cooling. In that instance, the processes of nucleation, growth (or spinodal decomposition) and coarsening are concurrent during aging of the initial structure [16].

In the first 3 h, the PS amplitudes for both the SC and FC samples evolve very fast (see Table 4). In Das et al. [5], it was shown that, for SDSS 2507 under real-service conditions, the embrittlement becomes pronounced when the amplitude is above ~25 at.%. Here, although the PS occurs under accelerated conditions, i.e., elevated temperature as compared to the service temperature, it is reasonable to suggest that the embrittlement occurs for both cooling conditions and is relevant for the service conditions. Considering the structural evolution embrittlement appears earlier in the SC sample than in the FC sample, as shown by the faster increase of amplitude. Hence, the slow cooling rate, corresponding to the thick plate scenario, would lead to a faster rate of PS and, therefore, more pronounced embrittlement in the real service conditions. Thus, in order to delay the embrittlement in the application of SDSS 2507, a thinner plate or faster cooling rate imposed by other means is to be preferred during the manufacturing process. The initial PS can in such cases be lowered, or even avoided [12], and thus the subsequent PS would be less pronounced.

## 5. Conclusions

In the present work, it is found that the cooling process after solution treatment imposed on SDSS 2507 has a distinct effect on the initial nanostructure, where decreasing the cooling rate after solution treatment will lead to a slight increase of the initial Cr amplitude. Further, this initial structure after slow cooling will then slightly affect the evolution of PS during subsequent aging at 475 °C, as shown by the larger wavelength and amplitude of PS: a faster cooling rate would lead to a slower rate of PS during aging. Therefore, in the industrial context, the PS can be delayed by increasing the cooling rate after solution treatment. This is normally achieved when thinner plates of DSS are manufactured.

**Author Contributions:** Conceptualization, J.L., Y.D. and P.H.; methodology, J.L., Y.D., S.M.K. and P.H.; software, J.L. and Y.D.; validation, J.L.; formal analysis, J.L., Y.D. and S.M.K.; investigation, J.L., Y.D., S.M.K.; resources, J.Y.J., S.W. and P.H.; data curation, J.L.; writing—original draft preparation, J.L.; writing—review and editing, J.L., Y.D., S.M.K., J.Y.J., S.W. and P.H.; visualization, J.L.; supervision, P.H.; project administration, J.L., S.W. and P.H.; funding acquisition, J.L., Y.D. and P.H. All authors have read and agreed to the published version of the manuscript.

**Funding:** This research was affiliated to project Cooler LSI, funded by VINNOVA (contract 2015–03453) within the Strategic Swedish Innovation Programme for Metallic Materials 2013–2016 and the Swedish industry; and the EIT RawMaterials project ENDUREIT (No. 18317). The research was funded by China Scholarship Council (CSC No. 201700260207).

**Institutional Review Board Statement:** Not applicable.

**Informed Consent Statement:** Not applicable.

**Data Availability Statement:** The SANS experiments data will be available to download via https://data.isis.stfc.ac.uk/doi/study/103212162 (accessed on 25 October 2021).

**Acknowledgments:** The authors acknowledge the award of beamtime (Experiment No. RB1910307, https://doi.org/10.5286/ISIS.E.RB1910307 (accessed on 25 October 2021)) at the STFC ISIS Pulsed Neutron & Muon Source, UK. Nataliya Limbach-Malyar is acknowledged for the assistance in the SANS beamtime. James Oliver is thanked for helping with the Gleeble heat treatments. J.L. thanks the China Scholarship Council (CSC No.201700260207) and Jernkontoret for financial support. Y.D. acknowledges receipt of an Olle Eriksson travel grant. This work benefited from the use of the SasView application, originally developed under NSF award DMR-0520547. SasView contains

code developed with funding from the European Union's Horizon 2020 research and innovation programme under the SINE2020 project, grant agreement No. 654000.

**Conflicts of Interest:** The authors declare no conflict of interest.

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
