# Peer review of "Effect of Cooling Rate after Solution Treatment on Subsequent Phase Separation Evolution in Super Duplex Stainless Steel 25Cr-7Ni (wt.%)"

_metals, doi:10.3390/met12050890_

Round 1

Reviewer 1 Report

The problem of phase separation caused by spinodal decomposition is a great problem in highly chromium alloyed steels operated around 475 ℃ temperature.

In many high chromium steels, spinodal decomposition occurs due to heat input. This is common for ferritic, duplex, and lean-duplex stainless steels and certain heat-resistant steels with high chromium content. It is known that for duplex stainless steels, spinodal decomposition occurs in the temperature range of approximately 250-500 ℃. In this process the ferrite is transformed into iron-rich and chromium-rich phases. The process is fastest close to 475 ℃ temperature, which corresponds to the nose point of the isothermal transformation diagram (TTT). Said phase transition dramatically changes the properties of the stainless steel. Steel becomes brittle and its corrosion resistance deteriorates. This degradation process is known in industrial practice as “475 ℃ embrittlement”.

The grain size of the phases produced by spinodal decomposition is in the 10-50 nm range therefore the investigation requires special techniques. The applied small-angle neutron scattering (SANS) is one of the most powerful investigation techniques for microstructure of spinodal alloys.

The work is a great study using SANS to study the process of spinodal decomposition. However, I do not see this conclusion sufficiently supported by the experimental results. The following statements of the conclusion are not convincing. “Further, this initial structure will then affect the evolution of PS during subsequent aging at 475 °C. At an early stage of PS, a faster cooling rate would decrease the kinetics of PS during aging.”

In Fig.5 the curves belonging to SC and FC states are very close to each other. On this basis, it cannot be reliably concluded that “At an early stage of PS, a SC.”

I suggest rethinking and amending the conclusions.

I also recommend checking the wording and the language for correctness. I give just one example here: “a faster cooling rate would decrease the kinetics of PS during aging”. The kinetics of a process do not decrease or increase. The rate of the process may decrease or increase.

Author Response

Reviewer 1: The problem of phase separation caused by spinodal decomposition is a great problem in highly chromium alloyed steels operated around 475 ℃ temperature.

In many high chromium steels, spinodal decomposition occurs due to heat input. This is common for ferritic, duplex, and lean-duplex stainless steels and certain heat-resistant steels with high chromium content. It is known that for duplex stainless steels, spinodal decomposition occurs in the temperature range of approximately 250-500 ℃. In this process the ferrite is transformed into iron-rich and chromium-rich phases. The process is fastest close to 475 ℃ temperature, which corresponds to the nose point of the isothermal transformation diagram (TTT). Said phase transition dramatically changes the properties of the stainless steel. Steel becomes brittle and its corrosion resistance deteriorates. This degradation process is known in industrial practice as “475 ℃ embrittlement”.

The grain size of the phases produced by spinodal decomposition is in the 10-50 nm range therefore the investigation requires special techniques. The applied small-angle neutron scattering (SANS) is one of the most powerful investigation techniques for microstructure of spinodal alloys.

Response: Thank you for your feedback. We hope we have addressed all your concerns in our revision.

Reviewer 1: The work is a great study using SANS to study the process of spinodal decomposition. However, I do not see this conclusion sufficiently supported by the experimental results. The following statements of the conclusion are not convincing. “Further, this initial structure will then affect the evolution of PS during subsequent aging at 475 °C. At an early stage of PS, a faster cooling rate would decrease the kinetics of PS during aging.”

In Fig.5 the curves belonging to SC and FC states are very close to each other. On this basis, it cannot be reliably concluded that “At an early stage of PS, a SC.”

I suggest rethinking and amending the conclusions.

Response: Thank you for your suggestion. Our conclusion is made mainly based on the SANS results, i.e. wavelength and amplitude in Table 3 and Table 4. Although the lines of SC and FC from Fig.5 are seems to be close, the difference is still visible. Those small difference correlates to the slight difference in Table 3 and Table 4. Thus, we have now revised it into: ‘Further, this initial structure will then slightly affect the evolution of PS during subsequent aging at 475 C, as shown by the larger wavelength and amplitude of PS: a faster cooling rate would lead to a slower rate of PS during aging.’

Reviewer 1: I also recommend checking the wording and the language for correctness. I give just one example here: “a faster cooling rate would decrease the kinetics of PS during aging”. The kinetics of a process do not decrease or increase. The rate of the process may decrease or increase.

Response: We thank the Reviewer for drawing this to our attention. We have amended our usage of ‘kinetics’ throughout the manuscript.

Reviewer 2 Report

I recommend submitting the manuscript "Effect of cooling rate after solution treatment on subsequent phase separation", I find it very good.

Author Response

Reviewer 2: I recommend submitting the manuscript "Effect of cooling rate after solution treatment on subsequent phase separation", I find it very good.

Response: We very much thank the Reviewer for their kind comments.

Reviewer 3 Report

In this manuscript, the authors studied the effect of the cooling rate of super duplex stainless steel 25Cr-7Ni (wt. %), and the effect of the initial structure on phase separation evolution during subsequent aging. The SANS method and data are well structured and the explanation of the results is, in general, in accordance with the data presented, although the amount of experimental data is quite small. The discussion and conclusions obtained from the information of the work are adequate but the current novelty of this paper is not sufficiently emphasized.  In my opinion, the manuscript entitled " Effect of cooling rate after solution treatment on subsequent phase separation evolution in super duplex stainless steel 25Cr- 7Ni (wt.%)” (authors: Jianling Liu, Yadunandan Das, Stephen M. King, Jan Y. Jonsson, Sten Wessman, Peter Hedström) is suitable for publication with minor corrections (Accept with minor revisions).

Author Response

We thank the Reviewer for their comments and apologise if the novelty of our paper was unclear. We have tried to emphasize the novelty in our revision in the Introduction as follows: ‘We describe the effect of cooling rate after solution treatment on the initial structure of the ferrite, and discuss the effect of the initial structure on the rate of PS during subsequent aging. In addition, suggestions to alleviate PS in the industrial production has been given’; and in Conclusion part as follows: ‘Further, this initial structure will then slightly affect the evolution of PS during subsequent aging at 475 C, as shown by the larger wavelength and amplitude of PS: a faster cooling rate would lead to a slower rate of PS during aging. Therefore, in the industrial context, the PS can be delayed by increasing the cooling rate after solution treatment. This may also be achieved by having a thinner plate during the manufacturing of DSS.’

Round 2

Reviewer 1 Report

The improved version of the paper can be published in its preset form.